# PARAMETER-EFFICIENT TRANSFER LEARNING WITH DIFF PRUNING

## ABSTRACT

While task-specific finetuning of deep networks pretrained with self-supervision has led to significant empirical advances in NLP, their large size makes the standard finetuning approach difficult to apply to multi-task, memory-constrained settings, as storing the full model parameters for each task become prohibitively expensive. We propose *diff pruning* as a simple approach to enable parameter-efficient transfer learning within the pretrain-finetune framework. This approach views finetuning as learning a task-specific "diff" vector that is applied on top of the pretrained parameter vector, which remains fixed and is shared across different tasks. The diff vector is adaptively pruned during training with a differentiable approximation to the $L_0$-norm penalty to encourage sparsity. Diff pruning becomes parameter-efficient as the number of tasks increases, as it requires storing only the nonzero positions and weights of the diff vector for each task, while the cost of storing the shared pretrained model remains constant. We find that models finetuned with diff pruning can match the performance of fully finetuned baselines on the GLUE benchmark while only modifying 0.5% of the pretrained model's parameters per task.

## 1 INTRODUCTION

Task-specific finetuning of pretrained deep networks has become the dominant paradigm in contemporary NLP, achieving state-of-the-art results across a suite of natural language understanding tasks (Devlin et al., 2019; Liu et al., 2019c; Yang et al., 2019; Lan et al., 2020). While straightforward and empirically effective, this approach is difficult to scale to multi-task, memory-constrained settings (e.g. for on-device applications), as it requires shipping and storing a full set of model parameters for each task. Inasmuch as these models are learning generalizable, task-agnostic language representations through self-supervised pretraining, finetuning the entire model for each task is an especially inefficient use of model parameters.

A popular approach to parameter-efficiency is to learn sparse models for each task where a subset of the final model parameters are exactly zero (Gordon et al., 2020; Sajjad et al., 2020; Zhao et al., 2020; Sanh et al., 2020). Such approaches often face a steep sparsity/performance tradeoff, and a substantial portion of nonzero parameters (e.g. 10%-30%) are still typically required to match the performance of the dense counterparts. An alternative is to use multi-task learning or feature-based transfer for more parameter-efficient transfer learning with pretrained models (Liu et al., 2019b; Clark et al., 2019; Stickland & Murray, 2019; Reimers & Gurevych, 2019; Feng et al., 2020). These methods learn only a small number of additional parameters (e.g. a linear layer) on top of a shared model. However, multi-task learning generally requires access to all tasks during training to prevent catastrophic forgetting (French, 1999), while feature-based transfer learning (e.g. based on task-agnostic sentence representations) is typically outperformed by full finetuning (Howard & Ruder, 2018).

Adapters (Rebuffi et al., 2018) have recently emerged as a promising approach to parameter-efficient transfer learning within the pretrain-finetune paradigm (Houlsby et al., 2019; Pfeiffer et al., 2020a;b;c). Adapter layers are smaller, task-specific modules that are inserted between layers of a pretrained model, which remains fixed and is shared across tasks. These approaches do not require access to all tasks during training making them attractive in settings where one hopes to obtain and share performant models as new tasks arrive in stream. Houlsby et al. (2019) find that adapter layers trained on BERT can match the performance of fully finetuned BERT on the GLUE benchmark (Wang et al., 2019a) while only requiring 3.6% additional parameters (on average) per task.

In this work, we consider a similar setting as adapters but propose a new *diff pruning* approach with the goal of even more parameter-efficient transfer learning. Diff pruning views finetuning as learning a task-specific difference vector that is applied on top of the pretrained parameter vector, which remains fixed and is shared across different tasks. In order to learn this vector, we reparameterize the task-specific model parameters as $\boldsymbol{\theta}_{\text{task}} = \boldsymbol{\theta}_{\text{pretrained}} + \boldsymbol{\delta}_{\text{task}}$, where the pretrained parameter vector $\boldsymbol{\theta}_{\text{pretrained}}$ is fixed and the task-specific diff vector $\boldsymbol{\delta}_{\text{task}}$ is finetuned. The diff vector is regularized with a differentiable approximation to the $L_0$-norm penalty (Louizos et al., 2018) to encourage sparsity. This approach can become parameter-efficient as the number of tasks increases as it only requires storing the nonzero positions and weights of the diff vector for each task. The cost of storing the shared pretrained model remains constant and is amortized across multiple tasks. On the GLUE benchmark (Wang et al., 2019a), diff pruning can match the performance of the fully finetuned BERT baselines while finetuning only $0.5\%$ of the pretrained parameters per task, making it a potential alternative to adapters for parameter-efficient transfer learning.

## 2 BACKGROUND: TRANSFER LEARNING FOR NLP

The field of NLP has recently seen remarkable progress through *transfer learning* with a pretrain-and-finetune paradigm, which initializes a subset of the model parameters for all tasks from a pretrained model and then finetunes on a task specific objective. Pretraining objectives include context prediction (Mikolov et al., 2013), autoencoding (Dai & Le, 2015), machine translation (McCann et al., 2017), and more recently, variants of language modeling (Peters et al., 2018; Radford et al., 2018; Devlin et al., 2019) objectives.

Here we consider applying transfer learning to multiple tasks. We consider a setting with a potentially unknown set of tasks, where each $\tau \in \mathcal{T}$ has an associated training set $\{x_\tau^{(n)}, y_\tau^{(n)}\}_{n=1}^N$.[1] For all tasks, the goal is to produce (possibly tied) model parameters $\boldsymbol{\theta}_\tau$ to minimize the empirical risk,

$$\min_{\boldsymbol{\theta}_\tau} \ \frac{1}{N} \sum_{n=1}^N \mathcal{L}\left(f(x_\tau^{(n)}; \boldsymbol{\theta}_\tau), y_\tau^{(n)}\right) + \lambda R(\boldsymbol{\theta}_\tau)$$

where $f(\cdot; \boldsymbol{\theta})$ is a parameterized function over the input (e.g. a neural network), $\mathcal{L}(\cdot, \cdot)$ is a loss function (e.g. cross-entropy), and $R(\cdot)$ is an optional regularizer with hyperparameter $\lambda$.

This multi-task setting can use the pretrain-then-finetune approach by simply learning independent parameters for each task; however the large size of pretrained models makes this approach exceedingly parameter inefficient. For example, widely-adopted models such as BERT$_{\text{BASE}}$ and BERT$_{\text{LARGE}}$ have 110M and 340M parameters respectively, while their contemporaries such as T5 (Raffel et al., 2020), Megatron-LM (Shoeybi et al., 2019), and Turing-NLG (Rajbhandari et al., 2019) have parameter counts in the billions. Storing the fully finetuned models becomes difficult even for a moderate number of tasks.[2] A classic approach to tackling this parameter-inefficiency (Caruana, 1997) is to train a single shared model (along with a task-specific output layer) against multiple tasks through joint training. However, the usual formulation of multi-task learning requires the set of tasks $\mathcal{T}$ to be known in advance in order to prevent catastrophic forgetting (French, 1999),[3] making it unsuitable for applications in which the set of tasks is unknown (e.g. when tasks arrive in stream).

## 3 DIFF PRUNING

Diff pruning formulates task-specific finetuning as learning a diff vector $\boldsymbol{\delta}_\tau$ that is added to the pretrained model parameters $\boldsymbol{\theta}_{\text{pretrained}}$. We first reparameterize the task-specific model parameters,

$$\boldsymbol{\theta}_\tau = \boldsymbol{\theta}_{\text{pretrained}} + \boldsymbol{\delta}_\tau,$$

---

[1]Therefore our setup is different from the classic multitask setting which usually assumes that set of tasks is known

[2]An intriguing line of work suggests that large-scale language models can be used *without* finetuning for a variety of tasks if given the appropriate context (Radford et al., 2019; Brown et al., 2020). While interesting, these models generally underperform task-specific models and require billions of parameters, though recent work suggests that they can be made substantially smaller (Schick & Schutze, 2020).

[3]However, work on *continual learning* mitigates these issues to an extent (Shin et al., 2017; Lopez-Paz & Ranzato, 2017; Lee et al., 2017; Kirkpatrick et al., 2017; Parisi et al., 2018).

which results in the following empirical risk minimization problem,

$$\min_{\boldsymbol{\delta}_\tau} \ \frac{1}{N} \sum_{n=1}^{N} \mathcal{L}\left(f(x_\tau^{(n)}; \boldsymbol{\theta}_{\text{pretrained}} + \boldsymbol{\delta}_\tau), y_\tau^{(n)}\right) + \lambda R(\boldsymbol{\theta}_{\text{pretrained}} + \boldsymbol{\delta}_\tau).$$

This trivial reparameterization is equivalent to the original formulation. Its benefit comes in the multi-task setting where the cost of storing the pretrained parameters $\boldsymbol{\theta}_{\text{pretrained}}$ is amortized across tasks, and the only marginal cost for new tasks is the diff vector. If we can regularize $\boldsymbol{\delta}_\tau$ to be sparse such that $\|\boldsymbol{\delta}_\tau\|_0 \ll \|\boldsymbol{\theta}_{\text{pretrained}}\|_0$, then this approach can become more parameter-efficient as the number of tasks increases. We can specify this goal with an $L_0$-norm penalty on the diff vector,

$$R(\boldsymbol{\theta}_{\text{pretrained}} + \boldsymbol{\delta}_\tau) = \|\boldsymbol{\delta}_\tau\|_0 = \sum_{i=1}^{d} \mathbb{1}\{\delta_{\tau,i} \neq 0\}.$$

### 3.1 DIFFERENTIABLE APPROXIMATION TO THE $L_0$-NORM

This regularizer is difficult to directly optimize as it is non-differentiable. In order to approximate this $L_0$ objective, we follow the standard approach for gradient-based learning with $L_0$ sparsity using a relaxed mask vector (Louizos et al., 2018). This approach involves relaxing a binary vector into continuous space, and then multiplying it with a dense weight vector to determine how much of the weight vector is applied during training. After training, the mask is deterministic and a large portion of the diff vector is true zero.

To apply this method we first decompose $\boldsymbol{\delta}_\tau$ into a binary mask vector multiplied with a dense vector,

$$\boldsymbol{\delta}_\tau = \mathbf{z}_\tau \odot \mathbf{w}_\tau, \qquad\qquad \mathbf{z}_\tau \in \{0,1\}^d, \mathbf{w}_\tau \in \mathbb{R}^d$$

We can now instead optimize an expectation with respect to $\mathbf{z}_\tau$, whose distribution $p(\mathbf{z}_\tau; \boldsymbol{\alpha}_\tau)$ is initially Bernoulli with parameters $\boldsymbol{\alpha}_\tau$,

$$\min_{\boldsymbol{\alpha}_\tau, \mathbf{w}_\tau} \ \mathbb{E}_{\mathbf{z}_\tau \sim p(\mathbf{z}_\tau; \boldsymbol{\alpha}_\tau)} \left[ \frac{1}{N} \sum_{n=1}^{N} \mathcal{L}\left(f(x_\tau^{(n)}; \boldsymbol{\theta}_{\text{pretrained}} + \mathbf{z}_\tau \odot \mathbf{w}_\tau,), y_\tau^{(n)}\right) + \lambda \|\boldsymbol{\delta}_\tau\|_0 \right].$$

This objective is still difficult in practice due to $\mathbf{z}_\tau$'s being discrete (which requires the score function gradient estimator), but the expectation provides some guidance for empirically effective relaxations. We follow prior work (Louizos et al., 2018; Wang et al., 2019b) and relax $\mathbf{z}_\tau$ into continuous space $[0,1]^d$ with a stretched Hard-Concrete distribution (Jang et al., 2017; Maddison et al., 2017), which allows for the use of pathwise gradient estimators. Specifically, $\mathbf{z}_\tau$ is now defined to be a deterministic and (sub)differentiable function of a sample $\mathbf{u}$ from a uniform distribution,

$$\mathbf{u} \sim U(\mathbf{0}, \mathbf{1}), \qquad\qquad \mathbf{s}_\tau = \sigma\left(\log \mathbf{u} - \log(1 - \mathbf{u}) + \boldsymbol{\alpha}_\tau\right),$$
$$\bar{\mathbf{s}}_\tau = \mathbf{s}_\tau \times (r - l) + l, \qquad\qquad \mathbf{z}_\tau = \min(\mathbf{1}, \max(\mathbf{0}, \bar{\mathbf{s}}_\tau)).$$

Here $l < 0$ and $r > 1$ are two constants used to stretch $\mathbf{s}_\tau$ into the interval $(l, r)^d$ before it is clamped to $[0,1]^d$ with the $\min(\mathbf{1}, \max(\mathbf{0}, \cdot))$ operation. In this case we have a differentiable closed-form expression for the expected $L_0$-norm,

$$\mathbb{E}\left[\|\boldsymbol{\delta}_\tau\|_0\right] = \sum_{i=1}^{d} \mathbb{E}\left[\mathbb{1}\{\mathbf{z}_{\tau,i} > 0\}\right] = \sum_{i=1}^{d} \sigma\left(\boldsymbol{\alpha}_{\tau,i} - \log \frac{-l}{r}\right).$$

Thus the final optimization problem is given by,

$$\min_{\boldsymbol{\alpha}_\tau, \mathbf{w}_\tau} \ \mathbb{E}_{\mathbf{u} \sim U[\mathbf{0},\mathbf{1}]} \left[ \frac{1}{N} \sum_{n=1}^{N} \mathcal{L}\left(f(x_\tau^{(n)}; \boldsymbol{\theta}_{\text{pretrained}} + \mathbf{z}_\tau \odot \mathbf{w}_\tau,), y_\tau^{(n)}\right) \right] + \lambda \sum_{i=1}^{d} \sigma\left(\boldsymbol{\alpha}_{\tau,i} - \log \frac{-l}{r}\right),$$

and we can now utilize pathwise gradient estimators to optimize the first term with respect to $\boldsymbol{\alpha}_\tau$ since the expectation no longer depends on it.[4] After training we obtain the final diff vector $\boldsymbol{\delta}_\tau$ by sampling $\mathbf{u}$ once to obtain $\mathbf{z}_\tau$ (which is not necessarily a binary vector but has a significant number of dimensions equal to exactly zero due to the clamping function), then setting $\boldsymbol{\delta}_\tau = \mathbf{z}_\tau \odot \mathbf{w}_\tau$.[5]

---

[4]To reduce notation clutter we subsume the parameters of the task-specific output layer, which is not pretrained, into $\boldsymbol{\theta}_{\text{pretrained}}$. We do not apply the $L_0$-norm penalty on these parameters during training.

[5]We found sampling once to work as well as more complicated alternatives (e.g. based on multiple samples).

### 3.2 $L_0$-BALL PROJECTION WITH MAGNITUDE PRUNING FOR SPARSITY CONTROL

Differentiable $L_0$ regularization provides a strong way to achieve high sparsity rate. However, it would be ideal to have more fine-grained control into the exact sparsity rate in the diff vector, especially considering applications which require specific parameter budgets. As $\lambda$ is just the Lagrangian multiplier for the constraint $\mathbb{E}\left[\|\boldsymbol{\delta}_\tau\|_0\right] < \eta$ for some $\eta$, this could be achieved in principle by searching over different values of $\lambda$. However we found it more efficient and empirically effective to achieve an exact sparsity rate by simply projecting onto the $L_0$-ball after training.

Specifically we use magnitude pruning on the diff vector $\boldsymbol{\delta}_\tau$ and target a sparsity rate $t\%$ by only keeping the top $t\% \times d$ values in $\boldsymbol{\delta}_\tau$.[6] Note that unlike standard magnitude pruning, this is based on the magnitude of the diff vector values and not the model parameters. As is usual in magnitude pruning, we found it important to further finetune $\boldsymbol{\delta}_\tau$ with the nonzero masks fixed to maintain good performance (Han et al., 2016). Since this type of parameter-efficiency through projection onto the $L_0$-ball can be applied without adaptive diff pruning,[7] such an approach will serve as one of our baselines in the empirical study.

### 3.3 STRUCTURED DIFF PRUNING

Diff pruning, as presented above, is architecture-agnostic and does not exploit the underlying model structure—each dimension of $\mathbf{z}_\tau$ is independent from one another. While this makes the approach potentially more flexible, we might expect to achieve better sparsity/performance tradeoff through a structured formulation which encourages active parameters to group together and other areas to be fully sparse. Motivated by this intuition, we first partition the parameter indices into $G$ groups $\{g(1), \dots, g(G)\}$ where $g(j)$ is a subset of parameter indices governed by group $g(j)$.[8] We then introduce a scalar $\mathbf{z}_\tau^j$ (with the associated parameter $\boldsymbol{\alpha}_\tau^j$) for each group $g(j)$, and decompose the task-specific parameter for index $i \in g(j)$ as $\boldsymbol{\delta}_{\tau,i}^j = \mathbf{z}_{\tau,i} \times \mathbf{z}_\tau^j \times \mathbf{w}_{\tau,i}$. The expected $L_0$-norm is then given by,

$$\mathbb{E}\left[\|\boldsymbol{\delta}_\tau\|_0\right] = \sum_{j=1}^{G} \sum_{i \in g(j)} \mathbb{E}\left[\mathbb{1}\{\mathbf{z}_{\tau,i} \cdot \mathbf{z}_\tau^g > 0\}\right] = \sum_{j=1}^{G} \sum_{i \in g(j)} \sigma\left(\boldsymbol{\alpha}_{\tau,i} - \log\frac{-l}{r}\right) \times \sigma\left(\boldsymbol{\alpha}_\tau^j - \log\frac{-l}{r}\right),$$

and we can train with gradient-based optimization as before.

## 4 EXPERIMENTS

### 4.1 MODEL AND DATASETS

For evaluation we use the GLUE benchmark (Wang et al., 2019b), a popular finetuning dataset. Following adapters (Houlsby et al., 2019), we test our approach on the following subset of the GLUE tasks: Multi-Genre Natural Language Inference (**MNLI**), where the goal is two predict whether the relationship between two sentences is entailment, contradiction, or neutral (we test on both $MNLI_m$ and $MNLI_{mm}$ which respectively tests on matched/mismatched domains); Quora Question Pairs (**QQP**), a classification task to predict whether two question are semantically equivalent; Question Natural Language Inference (**QNLI**), which must predict whether a sentence is a correct answer to the question; Stanford Sentiment Treebank (**SST-2**), a sentence classification task to predict the sentiment of movie reviews; Corpus of Linguistic Acceptability (**CoLA**), where the goal is predict whether a sentence is linguistically acceptable or not; Semantic Textual Similarity Benchmark (**STS-B**), which must predict a similarity rating between two sentences; Microsoft Research Paraphrase Corpus (**MRPC**), where the goal is to predict whether two sentences are semantically equivalent; Recognizing Textual Entailment (**RTE**), which must predict whether a second sentence is entailed by the first. For evaluation, the benchmark uses Matthew's correlation for CoLA, Spearman for STS-B, $F_1$ score for MRPC/QQC, and accuracy for MNLI/QNLI/SST-2/RTE.

---

[6]Wang et al. (2019b) show that it also is possible to inject such a constraint softly into the training objective by regularizing the expected model size towards a certain rate. However, since the constraint is soft this approach also makes it difficult to target an exact sparsity rate.

[7]Concretely, one can obtain $\boldsymbol{\theta}_\tau$ through usual finetuning, set $\boldsymbol{\delta}_\tau = \boldsymbol{\theta}_\tau - \boldsymbol{\theta}_{\text{pretrained}}$, and then apply magnitude pruning followed by additional finetuning on $\boldsymbol{\delta}_\tau$.

[8]While groups can be defined in various ways, we found that defining groups based on each matrix/bias vector of the pretrained model was simple and worked well enough.

| | Total params | New params per task | QNLI* | SST-2 | $MNLI_m$ | $MNLI_{mm}$ | CoLA | MRPC | STS-B | RTE | QQP | Avg |
|---|---|---|---|---|---|---|---|---|---|---|---|---|
| Full finetuning | $9.00\times$ | 100% | 91.1 | 94.9 | 86.7 | 85.9 | 60.5 | 89.3 | 87.6 | 70.1 | 72.1 | 80.9 |
| Adapters (8-256) | $1.32\times$ | 3.6% | 90.7 | 94.0 | 84.9 | 85.1 | 59.5 | 89.5 | 86.9 | 71.5 | 71.8 | 80.4 |
| Adapters (64) | $1.19\times$ | 2.1% | 91.4 | 94.2 | 85.3 | 84.6 | 56.9 | 89.6 | 87.3 | 68.6 | 71.8 | 79.8 |
| Full finetuning | $9.00\times$ | 100% | 93.4 | 94.1 | 86.7 | 86.0 | 59.6 | 88.9 | 86.6 | 71.2 | 71.7 | 80.6 |
| Last layer | $1.34\times$ | 3.8% | 79.8 | 91.6 | 71.4 | 72.9 | 40.2 | 80.1 | 67.3 | 58.6 | 63.3 | 68.2 |
| Non-adap. diff pruning | $1.05\times$ | 0.5% | 89.7 | 93.6 | 84.9 | 84.8 | 51.2 | 81.5 | 78.2 | 61.5 | 68.6 | 75.5 |
| Diff pruning | $1.05\times$ | 0.5% | 92.9 | 93.8 | 85.7 | 85.6 | 60.5 | 87.0 | 83.5 | 68.1 | 70.6 | 79.4 |
| Diff pruning (struct.) | $1.05\times$ | 0.5% | 93.3 | 94.1 | 86.4 | 86.0 | 61.1 | 89.7 | 86.0 | 70.6 | 71.1 | 80.6 |

**Table 1:** GLUE benchmark test server results with BERT$_{\text{LARGE}}$ models. (Top) Results with adapter bottleneck layers (brackets indicate the size of bottlenecks), taken from from Houlsby et al. (2019). (Bottom) Results from this work. *QNLI results are not directly comparable across the two works as the GLUE benchmark has updated the test set since then. To make our results comparable the average column is calculated without QNLI.

For all experiments, we use the BERT$_{\text{LARGE}}$ model from Devlin et al. (2019), which has 24 layers, 1024 hidden size, 16 attention heads, and 340M parameters. We use the Huggingface Transformer library (Wolf et al., 2019) to conduct our experiments.

## 4.2 BASELINES

We compare both structured and non-structured variants of diff pruning against the following baselines: **Full finetuning**, which fully finetunes BERT$_{\text{LARGE}}$ as usual; **Last layer finetuning**, which only finetunes the penultimate layer (along with the final output layer)[9]; **Adapters** from Houlsby et al. (2019), which train task-specific bottleneck layers between between each layer of a pretrained model, where parameter-efficiency can be controlled by varying the size of the bottleneck layers; and **Non-adaptive diff pruning**, which performs diff pruning just based on magnitude pruning (i.e., we obtain $\boldsymbol{\theta}_\tau$ through usual finetuning, set $\boldsymbol{\delta}_\tau = \boldsymbol{\theta}_\tau - \boldsymbol{\theta}_{\text{pretrained}}$, and then apply magnitude pruning followed by additional finetuning on $\boldsymbol{\delta}_\tau$). For diff pruning we set our target sparsity rate to 0.5% and investigate the effect of different target sparsity rates in section 5.1.

## 4.3 IMPLEMENTATION DETAILS AND HYPERPARAMETERS

Diff pruning introduces additional hyperparameters $l, r$ (for stretching the Hard-Concrete distribution) and $\lambda$ (for weighting the approximate $L_0$-norm penalty). We found $l = -1.5, r = 1.5, \lambda = 1.25 \times 10^{-7}$ to work well across all tasks. We also initialize the weight vector $\mathbf{w}_\tau$ to $\mathbf{0}$, and $\boldsymbol{\alpha}_\tau$ to a positive vector (we use $\mathbf{5}$) to encourage $\mathbf{z}_\tau$ to be close to $\mathbf{1}$ at the start of training. While we mainly experiment with BERT$_{\text{LARGE}}$ to compare against prior work with adapters (Houlsby et al., 2019), in preliminary experiments we found these hyperparameters to work for finetuning RoBERTa (Liu et al., 2019c) and XLNet (Yang et al., 2019) models as well.

For all tasks we use a learning rate of $1 \times 10^{-5}$ and perform a hyperparameter search over batch size $\in \{4, 6, 8, 10\}$ and the number of epochs $\in \{2, 3, 4, 5\}$.[10] However we found the default settings used for regular finetuning as suggested in the original BERT paper to work well for most tasks. Finetuning with the fixed mask after projecting onto the $L_0$-ball with magnitude pruning is done with a learning rate of $5 \times 10^{-5}$ for 3 or 5 epochs (3 epochs for QNLI, SST-2, MNLI-m, MNLI-mm, CoLA, QQP, 5 epochs for MRPC, STS-B, RTE). Grouping for the structured version of diff pruning is based on the matrix/bias vectors (i.e. parameters that belong to the same matrix or bias vector are assumed to be in the same group), which results in 393 groups.[11]

## 5 RESULTS AND ANALYSIS

Our main results on the GLUE benchmark are shown in Table 1. Structured diff pruning can match the performance of a fully finetuned BERT$_{\text{LARGE}}$ model while only requiring 0.5% additional pa-

---

[9]Wu et al. (2020) observe that finetuning later layers generally performs better than finetuning earlier layers

[10]For the larger QNLI, SST-2, MNLI-m, MNLI-mm, CoLA, QQP datasets, we use batch size of 8 over 3 epochs. For the smaller MRPC, STS-B, RTE datasets, we use batch size of 8 over 3 epochs.

[11]This definition of groups is implementation-specific since it depends on how one concatenates the input vector before each affine layer. Our grouping is based on Huggingface's BERT implementation at commit `656e1386a296d696327a9db37de2ccccc79e2cc7` (available at `https://github.com/huggingface/transformers/blob/656e1386a296d696327a9db37de2ccccc79e2cc7/src/transformers/modeling_bert.py`). In preliminary experiments we found this simple definition to work well compared to alternative group definitions (e.g. based on individual neurons).

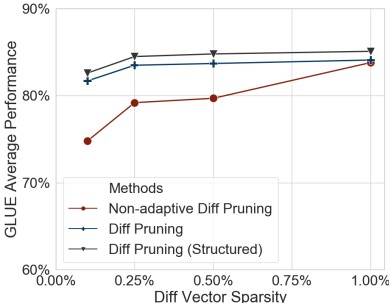

| | Pruned Diff Groups | | | |
| | Non-structured | | Structured | |
| | # | % | # | % |
| MRPC | 24 | 6.1 | 52 | 13.2 |
| STS-B | 25 | 6.4 | 48 | 12.2 |
| RTE | 28 | 7.1 | 50 | 12.7 |
| Avg | 25.7 | 6.5 | 50.0 | 12.7 |

**Figure 1:** (Left) Average performance on the GLUE validation set across different target sparsity rates for the different methods. (Right) Number of groups where all of the parameters in the group are fully zero for structured vs. non-structured diff pruning at 0.5% target sparsity. We group based on each matrix/bias vector, resulting in 393 groups in total.

| Diff vector target sparsity | QNLI | SST-2 | $MNLI_m$ | $MNLI_{mm}$ | CoLA | MRPC | STS-B | RTE | QQP | Avg |
|---|---|---|---|---|---|---|---|---|---|---|
| 0.10% | 92.7 | 93.3 | 85.6 | 85.9 | 58.0 | 87.4 | 86.3 | 68.6 | 85.2 | 82.5 |
| 0.25% | 93.2 | 94.2 | 86.2 | 86.5 | 63.3 | 90.9 | 88.4 | 71.5 | 86.1 | 84.5 |
| 0.50% | 93.4 | 94.2 | 86.4 | 86.9 | 63.5 | 91.3 | 89.5 | 71.5 | 86.6 | 84.8 |
| 1.00% | 93.3 | 94.2 | 86.4 | 87.0 | 66.3 | 91.4 | 89.9 | 71.1 | 86.6 | 85.1 |
| 100% | 93.5 | 94.1 | 86.5 | 87.1 | 62.8 | 91.9 | 89.8 | 71.8 | 87.6 | 85.0 |

**Table 2:** Structured diff pruning results on the validation set with different target sparsity rates. Average performance includes all 9 tasks.

rameters per task. Diff pruning without structured sparsity also performs well, though slightly worse than the structured approach. Non-adaptive diff pruning, which magnitude prunes the diff vector without learning the binary mask $\mathbf{z}_\tau$, performs significantly worse, indicating the importance of learning the masking vector. Compared to adapters, diff pruning obtains similar performance while requiring fewer parameters per task, making it a potential alternative for parameter-efficient transfer learning.[12] We now perform a series of analysis experiments on the validation set.

## 5.1 VARYING THE TARGET SPARSITY

In Figure 1 (left), we plot results on the GLUE validation set averaged across all tasks at target sparsity rates of $0.1\%, 0.25\%, 0.5\%, 1.0\%$ for the different baselines. Structured diff pruning consistently outperforms non-structured and and non-adaptive variants across different sparsity rates. The advantage of adaptive methods becomes more pronounced at extreme sparsity rates. In Table 2, we report the breakdown of accuracy of structured diff pruning across different tasks and sparsity rates, where we observe that different tasks have different sensitivity to target sparsity rates. This suggests that we can obtain even greater parameter-efficiency through targeting task-specific sparsity rates in the diff vector.

## 5.2 STRUCTURED VS. NON-STRUCTURED DIFF PRUNING

Structured diff pruning introduces an additional mask per group, which encourages pruning of entire groups. This is less restrictive than traditional group sparsity techniques that have been used with $L_0$-norm relaxations which force all parameters in a group to share the same mask (Louizos et al., 2018; Wang et al., 2019b). However we still expect entire groups to be pruned out more often in the structured case, which might bias the learning process towards either eliminating completely or clustering together nonzero diffs. In Figure 1 (right), we indeed find that structured diff pruning leads to finetuned models that are much more likely to leave entire groups unchanged from their pretrained values (zero diffs).

## 5.3 TASK-SPECIFIC SPARSITY

Different layers of pretrained models have argued to encode different information (Liu et al., 2019a; Tenney et al., 2019). Given that each task will likely recruit different kinds of language phenomena embedded in the hidden layers, we hypothesize that diff pruning will modify different parts of the

---

[12]However diff pruning incurs additional storage cost due to storing the nonzero positions of the diff vector.

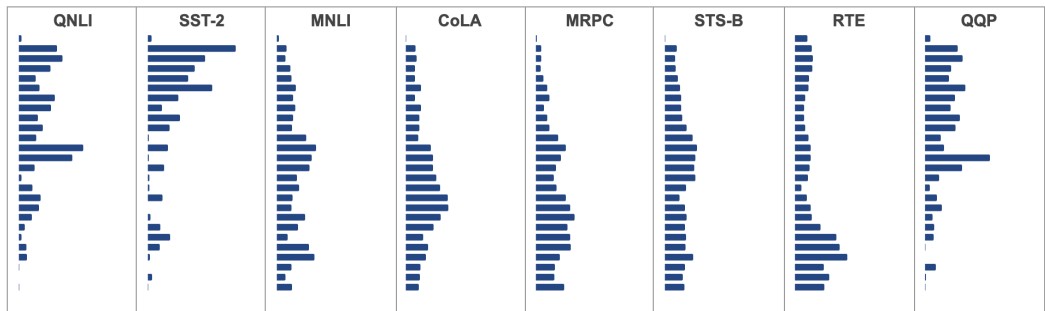

**Figure 2:** Percentage of modified parameters attributable to each layer for different tasks at 0.5% target sparsity. The layers are ordered from earlier to later (i.e. the embedding layer is shown at the top). The x-axis for each plot goes from 0% to 20%.

| | QNLI | SST-2 | $MNLI_m$ | $MNLI_{mm}$ | CoLA | MRPC | STS-B | RTE | QQP | Avg |
|---|---|---|---|---|---|---|---|---|---|---|
| Sparsity | 1.5% | 0.6% | 0.8% | 0.8% | 1.6% | 2.4% | 3.3% | 0.7% | 0.6% | 1.4% |
| Performance | 93.8 | 94.0 | 86.2 | 86.8 | 63.1 | 91.9 | 89.7 | 71.8 | 86.5 | 84.9 |
| With 0.5% sparsity | 93.4 | 94.2 | 86.4 | 86.9 | 63.5 | 91.3 | 89.5 | 71.5 | 86.6 | 84.8 |

**Table 3:** (Top) Sparsity and performance before magnitude pruning on the validation set with structured diff pruning. (Bottom) Performance with 0.5% target sparsity.

pretrained model through task-specific finetuning. Figure 2 shows the percentage of nonzero diff parameters attributable to the different layers for each task. We find that different tasks indeed modify different parts of the network, although there are some qualitative similarities between some tasks, for example between QNLI & QQP (both must encode questions), and MRPC & STS-B (both must predict similarity between sentences). The embedding layer is very sparsely modified for all tasks. While some of the variations in the sparsity distributions is due to simple randomness, we do observe some level of consistency over multiple runs of the same task, as shown in Figure 3 of the appendix.

The ability to modify different parts of the pretrained model for each task could explain the improved parameter-efficiency of our approach compared to Houlsby et al. (2019)'s adapter layers, which can only read/write to the pretrained model at certain points of the computational graph.[13] This potentially suggests that adapter layers with more fine-grained access into model internals (e.g. adapters for key/value/query transformations) might result in even greater parameter-efficiency. While left as future work, we also note that diff pruning can be applied in conjunction with adapters, which might further improve results.

## 5.4 Effect of $L_0$-ball Projection via Magnitude Pruning

Applying magnitude pruning to project onto the $L_0$-ball was crucial in achieving exact sparsity targets. As shown in Table 3, we observed little loss in performance through magnitude pruning. We re-iterate that it was crucial to finetune with the fixed mask in order to maintain good performance.[14]

## 5.5 SQuAD Extractive Question Answering

To demonstrate the effectiveness of our approach beyond classification, we additionally experiment on the extractive question answering task SQuAD, which asks model to select the answer span to a question given a Wikipedia paragraph. To make direct comparisons with Houlsby et al. (2019), we run all experiments on SQuAD v1.1. For diff pruning, we use the same general hyper-parameters as our full finetuning baseline.[15] Results are shown in Table 4. Diff pruning is able achieve comparable or better performance with only 1% additional parameters. Notably, we see that our method can improve the F1 score of full finetuning baseline by a significant margin (e.g. $90.8\% \Rightarrow 93.2\%$)

---

[13]To simulate this restricted setting, we tried applying diff pruning only on the dense transformations just before the output of each layer (i.e. after self-attention layers), and observed much worse performance.

[14]Even for the approach that does not apply magnitude pruning, we found it helpful to fix the mask $\mathbf{z}_\tau$ after an initial training phase and finetune just $\mathbf{w}_\tau$.

[15]https://huggingface.co/transformers/v2.5.1/examples.html

|  | Sparsity | F1 |
|---|---|---|
| Full finetuning | 100% | 90.7% |
| Adapters | 2% | 90.4% |
| Full finetuning | 100% | 90.8% |
| Diff pruning | 1% | 92.1% |
| Diff pruning (struct.) | 1% | 93.2% |

**Table 4:** SQuAD validation results with BERTLARGE model.

while modifying many fewer parameters (e.g., $100\% \Rightarrow 1\%$), which potentially implies that diff pruning can have a useful regularization effect.

## 6 DISCUSSION

### 6.1 MEMORY REQUIREMENTS

For training, our approach requires more memory than usual finetuning due to additionally optimizing $\boldsymbol{\alpha}_\tau$ and $\mathbf{w}_\tau$. This did not present a significant challenge for pretrained models that we experimented with in this study, since majority of GPU memory was utilized by the minibatch's activation layers. However, this could present an issue as model sizes get larger and larger. While training efficiency was not a primary concern of this work, diff pruning takes approxiamtely $1.5\times$ to $2\times$ more time per batch, which results in slower training.

After training, storing the task-specific diff vector requires storing a compressed version with both the nonzero positions and weights, which incurs additional storage requirements.

### 6.2 INFORMATION-EFFICIENT TRANSFER LEARNING

Efficiently representing pretrained models adapted to new tasks is becoming an increasingly important problem in contemporary NLP. This paper focuses on a rather narrow definition of efficiency—parameter-efficiency. An interesting direction might be to target generalizations of parameter-efficiency, for example, information-efficiency, which aims to minimize the number of bits required to represent the task-specific model when given the pretrained model for free. This view can suggest other avenues for achieving information-efficient transfer learning: for example, "what is the minimum number of (potentially synthetic) datapoints that we can finetune BERT on to obtain a good task-specific model?",[16] or "what is the shortest prefix string that we can condition GPT3 on for it to become a good task-specific model"?

## 7 RELATED WORK

**Multi-task learning**   Multi-task learning (Caruana, 1997), broadly construed, aims to learn models and representations that can be utilized across a diverse range of tasks, and offers a natural approach to training parameter-efficient deep models. Several works have shown that a single BERT model can obtain good performance across multiple tasks when jointly trained (Liu et al., 2019b; Clark et al., 2019; Stickland & Murray, 2019). Adapter layers, which are task-specific layers that read and write to layers of a shared model (Rebuffi et al., 2018), offer an alternative approach to multi-task learning that does not require access to all tasks during training, and have also been applied to obtain parameter-efficient BERT models (Houlsby et al., 2019; Pfeiffer et al., 2020a;b;c). A related line of work targets extreme parameter-efficiency through task-agnostic sentence representations that can be used without finetuning for downstream tasks (Le & Mikolov, 2014; Kiros et al., 2015; Wieting et al., 2016; Hill et al., 2016; Arora et al., 2017; Conneau et al., 2017; Cer et al., 2018; Zhang et al., 2018; Subramanian et al., 2018; Zhang et al., 2020). Reimers & Gurevych (2019), building on the earlier work of Conneau et al. (2017), show that BERT finetuned on natural language inference obtains sentence representations that perform well across multiple sentence-level tasks. These feature-based transfer learning methods are however generally outperformed by fully finetuned models (Howard & Ruder, 2018).

---

[16]Dataset distillation (Wang et al., 2018) tackles this question in the context of vision models.

**Model compression**   There has been much recent work on compressing pretrained trained with self-supervision (see Ganesh et al. (2020) for a recent survey). A particularly promising line of work focuses on obtaining smaller pretrained models (for subsequent finetuning) through weight pruning (Gordon et al., 2020; Sajjad et al., 2020; Chen et al., 2020) and/or knowledge distillation (Sanh et al., 2019; Sun et al., 2019; Turc et al., 2019; Jiao et al., 2019; Sun et al., 2020). It would be interesting to see whether our approach can be applied on top of these smaller pretrained models to for even greater parameter-efficiency.

**Learning to prune**   Our work is closely related to the line of work on learning to prune pretrained models with differentiable relaxations of binary masks (Wang et al., 2019b; Zhao et al., 2020; Sanh et al., 2020; Radiya-Dixit & Wang, 2020). While these works also enable parameter-efficient transfer learning, they generally apply the masks directly on the pretrained parameters instead of on the difference vector as in the present work.

**Regularization towards pretrained models**   Finally, diff pruning is also related to works which regularize the learning process towards pretrained models for continual learning (Kirkpatrick et al., 2017; Schwarz et al., 2018), domain adaptation (Wiese et al., 2017; Miceli Barone et al., 2017), and stable finetuning (Lee et al., 2020). These works typically do not utilize sparse regularizers and target a different goal than parameter-efficiency.

## 8   CONCLUSION

We propose diff pruning as a simple approach for parameter-efficient transfer learning with pretrained models. Experiments on standard NLP benchmarks and models show that diff pruning can match the performance of fully finetuned baselines while requiring only a few additional parameters per task. We also propose a structured variant of diff pruning which provides further improvements. Future work will consider (i) applying this approach to other architectures (e.g. ConvNets for vision applications), (ii) injecting parameter-efficiency objectives directly into the pretraining process (to pretrain models that are better suited towards sparse transfer learning), and (iii) combining diff pruning with other techniques (e.g. adapters) to achieve even greater parameter-efficiency.

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

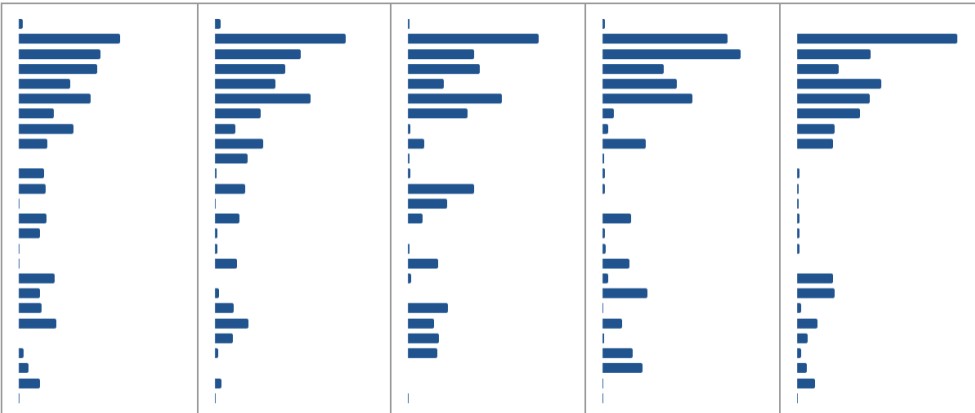

**Figure 3:** Percentage of modified parameters attributable to each layer for 5 different runs of SST-2 at 0.5% target sparsity. The layers are ordered from earlier to later (i.e. the embedding layer is shown at the top). The x-axis for each plot goes from 0% to 20%.

Zhilin Yang, Zihang Dai, Yiming Yang, Jaime Carbonell, Russ R Salakhutdinov, and Quoc V Le. XLNet: Generalized Autoregressive Pretraining for Language Understanding. In *Proceedings of NeurIPS*, 2019.

Minghua Zhang, Yunfang Wu, Weikang Li, and Wei Li. Learning universal sentence representations with mean-max attention autoencoder. In *Proceedings of EMNLP*, 2018.

Yan Zhang, Ruidan He, Zuozhu Liu, Kwan Hui Lim, and Lidong Bing. An Unsupervised Sentence Embedding Method byMutual Information Maximization. In *Proceedings of EMNLP*, 2020.

Mengjie Zhao, Tao Lin, Martin Jaggi, and Hinrich Schutze. Masking as an Efficient Alternative to Finetuning for Pretrained Language Models. *arXiv:2004.12406*, 2020.

# A    APPENDIX

## A.1    CONSISTENCY OF NONZERO PARAMETERS

Figure 3 shows the percentage of modified parameters attributable to each layer across 5 runs of SST-2. We find that there is nonotrivial variation in sparsity across runs, but also a degree of consistency. For example, the first layer is modified considerably more than other layers across all runs.

