# OpenReview forum: "Parameter-Efficient Transfer Learning with Diff Pruning"
_ICLR.cc/2021/Conference — Reject_

### Official Review · AnonReviewer4 · 2020-10-25

**Rating:** 8
**Confidence:** 4

**Review:**

This paper proposes diff pruning, an alternative paradigm for parameter-efficient transfer learning of pre-trained models. Similar to adapters, diff pruning leaves the body of the pre-trained model unchanged. Rather than inserting additional task-specific parameters into the pre-trained model, diff pruning adds reparameterizes the parameters of the transferred model $\theta_\tau$ by adding a diff vector $\delta_\tau$ to them: $\theta_\tau = \theta_{\text{pretrained}} + \delta_\tau$. Parameter efficiency is achieved by regularizing $\theta_\tau$ to be sparse. The authors achieve this by using a relaxed mask vector to approximate the $L_0$ norm. They also propose a way to control for a specific sparsity rate via projection onto the $L_0$ ball after training and to enforce group sparsity that takes the model's structure into account. The approach is evaluated on the GLUE benchmark where it achieves competitive performance to full fine-tuning a BERT Large model and adapters while being more parameter-efficient than both of them.

Pros:
1. The proposed method is intuitive and the different modelling choices are principled and well motivated.
2. The method achieves strong results. It is competitive with full fine-tuning and more parameter-efficient than adapters, the prevalent approach for parameter-efficient transfer learning.
3. The authors show how to effectively control the sparsity rate and incorporating structure via group sparsity brings further gains.
4. The authors conduct extensive analyses, which pre-empted many of my questions, such as the variation across different sparsity masks, sparsity patterns across different tasks, etc. Overall, the analyses shed additional light on the characteristics and preferences of different tasks in transfer learning.

Cons:
1. The approach is potentially more complicated than the baseline, so it is important that the authors open-source their code.
2. The diff vector is distributed over the entire set of parameters of the model rather than focused in a few layers. This makes it potentially harder to combine the diff vectors from different tasks as can be done with adapters (see e.g. https://arxiv.org/abs/2005.00247) and to compose multiple diff vectors.

Questions:
1. Does a visualization of diff vectors of different tasks (such as using t-SNE) reveal any interesting patterns?
2. Are there any transfer settings that the addition of task-specific parameters can model but inserting layer-specific transformations via adapters cannot (or vice versa)? Adapters have been used to transfer across modalities such as languages (see e.g. Pfeiffer et al. (2020), https://arxiv.org/abs/2005.00052) and I am wondering whether the same would be possible by adding task-specific parameters.
3. How long does your approach take to converge in comparison to the baselines? How much longer do you need to fine-tune with non-zero masks for magnitude pruning for sparsity control? What is the performance benefit of this further fine-tuning?

---

> ### Author Response · Authors · 2020-11-11
> **Review response**
>
> Thanks for the review! Please find the responses to various points below:
>
> - **".. so it is important that the authors open-source their code."** Yes, we will definitely be open-sourcing our code :)
>
> - **"Does a visualization of diff vectors of different tasks (such as using t-SNE) reveal any interesting patterns?"** In Figure 2 we find that visualization of mask distributions does reveal some interesting patterns. For example the mask distribution is similar between QNLI & QQP, which both deal with question inputs, and MRPC & STS-B, which both predict similarity between sentences. Visualization via t-SNE may be a bit difficult to apply in our case due the the dimensionality of the parameter space and the fact that we only have a few data points (i.e. one diff vector for each task). But we will explore other visualizations (e.g. based on cosine similarity) to see if any interesting patterns emerge.
>
> - **"Are there any transfer settings that the addition of task-specific parameters can model but inserting layer-specific transformations via adapters cannot (or vice versa)?"** This is a very interesting question. We hypothesize that the adapter layers proposed by Houlsby et al. (2019) are potentially more restrictive than full diff pruning since they can only read/write to certain parts of the model. Contemporaneous work suggests that more "fine-grained" adapters that have greater access to model internals can potentially improve results (https://openreview.net/forum?id=de11dbHzAMF). On the other hand, as noted by the reviewer diff pruning is potentially more restrictive than adapters when we consider applications where we must combine different modalities, or compose different adapter modules (e.g. if the tasks are related). An interesting future direction could thus involve *combining* both these approaches (i.e. learn task-specific adapter layers, but also learn task-specific diff vectors which can directly modify model parameters).
>
> - **"How long does your approach take to converge in comparison to the baselines?"** Our initial training stage (before finetuning with fixed non-zero masks) takes 3 epochs, which is same as the default number of epochs for BERT finetuning on GLUE from the original paper. However, our method does require additional training due to finetuning with fixed non-zero masks (as discussed below).
>
> -  **"How much longer do you need to fine-tune with non-zero masks for magnitude pruning for sparsity control? What is the performance benefit of this further fine-tuning?"** We finetune for 3-5 additional epochs (3 epochs for the larger QNLI, SST-2, MNLI-m, MNLI-mm, CoLA, QQP datasets, and 5 epochs for the smaller MRPC, RTE, STS-B datasets).  As has been observed in the  pruning literature (Han et al. 2015, See et al. 2016) we found this to be quite important for some tasks: for example MNLI performance goes from 84.3 to 86.9, while MRPC performance goes from 87.7 to 91.3. We will add these numbers to the paper.

---

### Official Review · AnonReviewer3 · 2020-10-27
**Interesting approach to reduce task specific parameters in transfer learning**

**Rating:** 6
**Confidence:** 4

**Review:**

The authors present an interesting approach to learn task-specific models with only a few tunable parameters. They propose learning a diff vector with a sparsity constraint and then pruning the vector using magnitude pruning. They also impose a structured sparsity constraint by introducing a group penalty.

This work is interesting and important with the growing size of the pretrained models. It enables the model to learn a new task with relatively few parameters would be very beneficial. The experiments and analysis conducted by the authors is thorough.

A few questions/thoughts which could improve the paper:
* It would be good to list the epochs for training for each of the different approaches? Authors mention 2,3,4,5 epochs for training but would good to highlight the cost or savings of fine tuning with their approach.
* In table 1, it would be also helpful to include some information about the finetuning steps/sec with the different approaches. This would help understand the tradeoff of memory vs compute for the proposed approach.
* What’s the intuition for structured pruning performing better than non unstructured? Typically, enforcing some structure should have worse accuracy. If the structure seems to help, then the model should be able to learn structure without the group constraint as well.

Overall, I recommend accepting the paper. As the size of pre-trained models is growing quite rapidly, research that investigates parameter sharing and adapting a pretrained model to a new task with few parameters is essential.

---

> ### Author Response · Authors · 2020-11-20
> **Review response**
>
> Thanks for your review! Please find our responses below
>
> - **"It would be good to list the epochs for training for each of the different approaches? Authors mention 2,3,4,5 epochs for training but would good to highlight the cost or savings of fine tuning with their approach."** For all tasks we initially train for 3 epochs (before finetuning with a fixed mask). Therefore our initial training  is comparable to standard BERT finetuning, which trains for 3 epochs by default. However, we do incur additional training time (3 additional epochs for most datasets except much smaller datasets such RTE) due to finetuning with a fixed mask (section 4.3). We also note that other approaches to parameter-efficient learning also perform some tuning over number of epochs and may take longer to converge. For examples, adapters tune over {3,20} epochs , as mentioned in section 3.2 of https://arxiv.org/pdf/1902.00751.pdf. As diff pruning targets parameter-efficiency rather than convergence speed, we didn't explore methods to train faster, but we nonetheless agree with the reviewer that having these numbers in the paper will be helpful and have included them in the newer version of the paper (footnote 10).
>
> - **"In table 1, it would be also helpful to include some information about the finetuning steps/sec with the different approaches. This would help understand the tradeoff of memory vs compute for the proposed approach."** We found diff pruning to be approximately `1.5~2 times to slower than full finetuning. Some of the overhead is due to our potentially suboptimal implementation (e.g. working with parameter blocks at the layer level instead of fully vectorized parameters), as training speed was not a primary concern of the work. We are currently exploring ways to make the implementation more efficient for practical deployment. We have added more discussion around this point in the rebuttal version of the paper in section 6.1. We also plan to open-source our code.
>
> - **What’s the intuition for structured pruning performing better than non unstructured? Typically, enforcing some structure should have worse accuracy. If the structure seems to help, then the model should be able to learn structure without the group constraint as well.** This is a good question. We think that structured pruning is providing a useful inductive bias to encourage parameters in similar groups to be active together, while encouraging other groups to be fully sparse. This is shown in Figure 1 (right), where we observe that that structured pruning indeed results in more groups that are fully zero. While theoretically the model should be able to learn the structure without the group constraint (as noted by the reviewer), in practice it may be useful to imbue these constraints into the model via our factorization. We observe that these types of inductive biases are common (e.g. Fully connected MLPs should theoretically be able to learn the structure of convolutional layers since MLPs are more expressive, but in practice imposing structural biases through convolutional layers is crucial to good performance in finite-data regimes).

---

### Official Review · AnonReviewer2 · 2020-10-27
**Interesting results, but better motivation and more experiments can further improve**

**Rating:** 5
**Confidence:** 4

**Review:**

This work combines model pruning with transfer learning/multi-task learning in NLP. Instead of finetuning pretrained models on each individual task, the author propose to learn 'residual' parameters with sparse masks for each task independently, hence reducing per-task parameter requirements. The evaluation on GLUE shows some promising results with better efficiency compared to adapter networks. However, I am concerned with the motivation and applicability of the proposed method, such that I am left with the impression that the method may be hard to use in practice.

Pros:
1. To the best of my knowledge, combining pruning on pretrained models to transfer learning in NLP is novel.
2. Empirical results show the method can performance comparably with adapter networks but with fewer parameters.

Cons (see below for detailed questions):
1. Although the method is proposed for 'multi-task' learning, it is not really evaluated on multi-task settings. The motivation and applicability are not very clear to me.
2. Compared to adapter networks, the proposed method contains more hyper-parameters and is harder to tune in practice.

Questions:
1. In section 2, the author introduced multi-task learning as the background. However, the proposed method is not really designed for multi-task learning (training multiple tasks simultaneously), but rather for transfer learning on multiple tasks independently. In practice, however, training some tasks together can improve performance (GLUE for example), but the proposed method is trained for each task independently and thus there is no positive transfer between tasks. So have you evaluated your method when trained multiple tasks simultaneously (where the shared parameters are jointly finetuned)? How does that work? This is an important extension that might be useful for the community.
2. Please point it out if I miss this information, but how did you select your hyper-parameters for your approach? Besides, how sensitive they are compared to adapter networks?
3. Prior work [1] has shown the resulting mask of pruning can be compared to evaluate task similarities. Did you have similar observations in your experiments?

Missing reference:
The idea of adding 'residual' parameters for new tasks is not new in lifelong learning. It is good to mention [2] and [3]. [1] is also related of comparing pruning masks for pretrained models.

Typo:
Page 4, last line: QQC -> QQP

[1] On negative interference in multilingual models: findings and a meta-learning treatment. Wang et al., EMNLP 2020.

[2] Progressive neural networks. Rusu et al., arxiv 2016

[3] BatchEnsemble: an alternative approach to efficient ensemble and lifelong learning. Wen et al., ICLR 2020.

---

> ### Author Response · Authors · 2020-11-10
> **Review response**
>
> Thanks for your review! Please find our responses below.
>
> - **"Although the method is proposed for 'multi-task' learning, it is not really evaluated on multi-task settings. The motivation and applicability are not very clear to me."**
> We agree that our use of "multi-task learning" in the background section is confusing and thank the reviewer for pointing this out. By multi-task we were referring to the fact that we want to adapt the pretrained model across multiple tasks with as few task-specific parameters as possible as tasks continuously arrive in stream, where we do not necessarily have access to previous tasks for joint training. This setup is crucially different from the conventional multi-task learning setting which usually assumes access to all tasks during training (e.g. https://arxiv.org/pdf/1901.11504.pdf, https://arxiv.org/abs/1907.04829). Our work is primarily concerned with the "adapter" setting of Houlsby et al. (2019) where we assume tasks arrive in stream and/or the full set of tasks is unknown. To avoid confusion, we have removied "multi-task learning" from the background section title and have clarified our setup more in this section.
>
> - **"Compared to adapter networks, the proposed method contains more hyper-parameters and is harder to tune in practice."** We are not sure we agree. As mentioned in 4.3, diff pruning introduces three additional hyperparameters, $l$, $r$ (for clamping the stretched Hard-Concrete sample) and $\lambda$ (for weighing the approximate $l_0$-norm penalty). Importantly, we use the *same* $l$, $r$, $\lambda$ values for all tasks (i.e. $\lambda = 1.25 \times 10^{-7}, l = -1.5, r=1.5$). These  values were found via a light grid search on the SST-2 dataset, which generalized well to across all datasets, demonstrating the robustness of these hyperparameters. Note that adapters also introduce various hyperparameters/modeling choices (e.g. size of hidden layers, parameterization of adapter modules) that are not present in our system.
>
> - **"So have you evaluated your method when trained multiple tasks simultaneously (where the shared parameters are jointly finetuned)? How does that work? This is an important extension that might be useful for the community."** Thanks for the suggestion. We think that this is an interesting future direction, but outside of the scope of this paper, especially given that multi-task learning often requires additional tweaks to make it work well (e.g. scheduled/weighted sampling of tasks during training). Our work is not in the traditional multi-task setting (e.g. as in Clark et al. 2019, Liu et al. 2019, Stickland and Murray 2019), but rather in the "adapter" setting of Houlsby et al. 2019, which targets a different goal. We also observe that even when it is possible to train jointly, multitask learning also may not be always ideal due to "negative transfer" (https://arxiv.org/pdf/2005.00944.pdf). We will add clearer discussion around this in the next iteration of the paper.
>
> - **"Please point it out if I miss this information, but how did you select your hyper-parameters for your approach?** We selected the hyperparameters based on performance on the development set. In particular, $l$, $r$, $\lambda$  hyperparameters were found via just searching on the SST-2 dev set. For the other hyperparameters, we used the default BERT finetuning hyperparameters for all tasks, except for the three smaller datasets (RTE, STS-B, MRPC) where we tuned over epochs/batch size.
>
> - **"Besides, how sensitive they are compared to adapter networks?"** We found diff pruning to work well with the default BERT fine-tuning hyperparameters for most tasks, although as mentioned above, the three smaller datasets (RTE, STS-B, MRPC)  datasets did require small tuning (as has been widely noted in the BERT finetuning literature). We note that adapters perform a more extensive hyperparameter tuning over a wider range of values (see section 3.2 of the Houlsby et al. 2019), and further tune of adapter sizes to each task to obtain the best results (row 2 of Table 1). Further, as noted in Houlsby et al. 2019, the best adapter results are obtained by running the model on each task 5 times and picking the best performing model. In contrast we use the same diff pruning parameters for all tasks, and run each of our models once for each task, which again indicates the robustness of our approach.
>
> - **"Prior work [1] has shown the resulting mask of pruning can be compared to evaluate task similarities. Did you have similar observations in your experiments?"** Interesting question! Please see Figure 2, where we do indeed observe similarities across tasks based on the resulting mask in Figure 2 (e.g. QNLI and QQP have similar mask distributions presumably because they both deal with question inputs). But we will explore this idea more in the next iteration of the paper.
>
> - **"Missing reference...".** Thanks for these references! We will add them to the paper.

---

### Official Review · AnonReviewer1 · 2020-10-29
**Concern about problem setting**

**Rating:** 4
**Confidence:** 4

**Review:**

This work studies the problem of parameter-efficient transfer learning in the paradigm of pretraining/finetuning. The proposed method, diff pruning, can match the performance of fully finetuned baselines on the GLUE benchmark while only modifying 0.5 of the pretrained model's parameters per task.

I don't get the problem definition of this work. In other words, the authors need to better motivate and justify parameter efficiency. Given that all the parameters of a big model are used in downstream tasks, what are the benefits of only modifying a few parameters? Here are several possibilities.
1. Does it speed up the finetuning process by modifying a few parameters?
2. Does it speed up inference for downstream tasks?
3. Another possibility is model size reduction, which leads to reduced storage costs. "This approach can become parameter-efficient as the number of tasks increases as it only requires storing the nonzero positions and weights of the diff vector for each task. The cost of storing the shared pretrained model remains constant and is amortized across multiple tasks." Unfortunately, in real-world scenarios, inference of  large models like BERT_base/large is usually conducted on servers or in cloud, where storage is not a big issue while latency (speed) is critical. Inference in edge devices like phones, IoT sensors cares about storage, speed, and power consumption.

---

> ### Author Response · Authors · 2020-11-10
> **Parameter-efficiency justification**
>
> Thanks for your review! The main justification for parameter efficiency is indeed reduced storage costs for multiple tasks.
>
> The reviewer states that this is not really relevant because:
> "Unfortunately, in real-world scenarios, inference of large models like BERT_base/large is usually conducted on servers or in cloud, where storage is not a big issue while latency (speed) is critical."
> We certainly agree that this statement is roughly true *for now*, but such a characterization risks overfitting to the current state of affairs. In particular:
> - While storage may not be a concern for BERT-sized models on servers/cloud, these models are only getting larger and larger! So even for servers/cloud storage-efficiency may eventually become a concern.
> - Following much work on edge device implementations of ConvNets, there has also been recent work on edge device implementations of Transformer networks (see, e.g., https://arxiv.org/pdf/2005.14187.pdf).
> It is therefore not too difficult imagine a world where (for example) your phone has a fast hardware implementation of a Transformer network, and to use this across multiple tasks, one just needs to "swap out" the parameters specific to different tasks. In this case, storage efficiency becomes crucial, as it becomes impractical to store the full BERT parameters for each task.
> - Yet another scenario in which parameter-efficiency is relevant is when we consider *sharing* task-specific models (e.g. you want to distribute an update to your model but you already know that your users have access to the previous (or pretrained) versions of the model).
> - Finally, while parameter-efficient transfer learning is still somewhat of a niche area in NLP, we note that there has been increasing interest from the research community, as evidenced by the recent vibrant work on adapter-based approaches to achieving parameter-efficiency in NLP (e.g. see https://adapterhub.ml/). There has also been much work on parameter-efficient transfer learning in the vision community (e.g. see papers citing https://arxiv.org/abs/1803.10082).
>
> Given the above, we strongly agree with R3's comment: **"As the size of pre-trained models is growing quite rapidly, research that investigates parameter sharing and adapting a pretrained model to a new task with few parameters is essential."**
>
> Finally, while the main goal of the paper is parameter-efficiency, we observe that for some tasks, diff pruning has a useful regularization effect and actually *improves* over the fully finetuned models. For example in Table 2 we find that diff pruning improves performance on CoLA from 62.8 to 66.3. In subsequent experiments on SQuAD, we found diff pruning to improve results  from 90.8 to 93.2 with 1% sparsity.
>
> We nonetheless thank the reviewer for surfacing these concerns and will make sure to emphasize these points more in the next iteration of the paper.

---

> > ### Comment · AnonReviewer1 · 2020-11-23
> > **Concerns about problem setting**
> >
> > Thanks for the clarification, which unfortunately leads to more questions.
> >
> > 1. "The main justification for parameter efficiency is indeed reduced storage costs for multiple tasks." If the goal is to reduce storage cost, a more straightforward approach is model compression. There are many recently proposed methods, including TinyBERT, DynaBERT, etc. Those methods achieve comparable accuracy as BERT using only 15-25% parameters. Note that the proposed method in this work still needs to use the same number of parameters as original BERT in inference.
> >
> > 2. "While storage may not be a concern for BERT-sized models on servers/cloud, these models are only getting larger and larger! So even for servers/cloud storage-efficiency may eventually become a concern." Indeed if models get larger and larger, they will eventually become a concern; however, the concern is computational cost rather than storage cost. For example, GPT-3 has 175 billion parameters, which is not challenging for cloud/server-side storage, but challenging for real-time inference, even in the cloud.
> >
> > 3. In terms of edge devices, the method proposed in this work cannot be deployed in edge devices, as the total number of parameters and computational cost of the model are not reduced.
> >
> > Overall, those justifications don't really "justify" the problem setting.

---

> > > ### Author Response · Authors · 2020-11-23
> > > **Response**
> > >
> > > Thanks for responding!
> > >
> > > - **"There are many recently proposed methods, including TinyBERT, DynaBERT, etc. Those methods achieve comparable accuracy as BERT using only 15-25% parameters."** This is not true? There is nontrivial drop in accuracy using these methods (see Table 1 of https://arxiv.org/pdf/1909.10351.pdf and Table 1 of https://arxiv.org/pdf/2004.04037.pdf). Also, even if these methods reach comparable accuracy with 15% parameters (which, again, is just not true), diff pruning is actually *more* parameter efficient when we consider the total number of parameters for the 9 GLUE tasks! In particular, 15% compression would imply 15%$\times$9 = 1.35, i.e. 1.35 times the total number of parameters, whereas with diff pruning we incur (1 + 0.005$\times$9) = 1.045, i.e. 1.045 times the total number of parameters. As the number of tasks increases, diff pruning will become even more parameter efficient.
> > >
> > > - **For example, GPT-3 has 175 billion parameters, which is not challenging for cloud/server-side storage, but challenging for real-time inference, even in the cloud.** This assumes that these models will always be stored on the cloud/server-side. Who is to say that next generation laptops won't ship with some version of these models? Could we have predicted thirty years ago that cell phones would ship with built-in ConvNets? Just because something is done a certain way now does not mean it will be done that way forever.
> > >
> > > - **In terms of edge devices, the method proposed in this work cannot be deployed in edge devices, as the total number of parameters and computational cost of the model are not reduced.** Again, this is only true for now. Edge device implementations will likely require combination of several techniques for deployment (e.g. distillation based approaches combined with methods for parameter efficiency).
> > >
> > > We kindly ask the reviewer to consider reviewing the contents of the paper rather than dismissing the paper outright just based on the problem setup. Thanks!

---

### Author Response · Authors · 2020-11-22
**Updated paper version**

We want to thank all the reviewers for their thoughtful comments. We have responded to the reviewer-specific points in our individual rebuttals.

We have also updated the paper with additional experiments on SQuAD (Table 4 in the updated paper), where we observe that diff pruning:

- generalizes well to SQuAD, which shows that it can be applied to tasks outside of GLUE
- can have a regularization effect and actually *improve* over full finetuning. In particular, we obtain the following F1 numbers on SQuAD v1.1 validation set :

Full-finetuning: 90.8
Adapters (2% sparsity): 90.4
Diff pruning (1% sparsity): 93.2

(Adapters results are from section 3.5 of Houlsby et al. 2019)

Finally, per R2's comments we have removed "multi-task learning" in the background section title and have clarified the setting of this paper.

---

### Decision · Program_Chairs · 2021-01-07
**Final Decision**

**Decision:**

Reject

**Comment:**

This paper studies a problem setup of parameter-efficient transfer learning for large-scale deep models. The approach consists of learning a diff vector with a sparsity constraint and then pruning the vector using magnitude pruning. A group penalty is also introduced to enhance structured sparsity. The main motivation is that for each new task, we only need to add a few parameters based on a pre-trained model without fine-tuning it.

The proposed approach possesses technical soundness and shows empirical efficacy for the studied problem setup. During the rebuttal and discussion phases, two of the reviewers raised two major concerns based on which they strongly disagreed with acceptance:
- The problem setup is not elaborated sufficiently and falls short of plausibility. An approach targeting at efficiency should either improve inference speed or reduce storage cost. Unfortunately, neither advantage has been well approached.
- The technical novelty is somewhat incremental, given the rich previous work on residual adapter, network re-parameterization, and network compression (pruning, sparsity etc.).

AC read the paper and agreed that, while the paper has some merit such as a better model for the particular problem setup, the reviewers' concerns are reasonable and need to be addressed in a more convincing way. For example, try to study a practical application in which the proposed approach is essential and useful for efficiency enhancement.